

# Heterogeneous future Arctic Ocean primary productivity changes projected in CMIP6

Léna L.Champiot-Bayard[1], Lester L.Kwiatkowski[1], Martin M.Vancoppenolle[1]

[1] LOCEAN Laboratory, Sorbonne Université-CNRS-IRD-MNHN, Paris, 75005, France

*Correspondence to*: Léna Champiot-Bayard (lena.champiot-bayard@locean.ipsl.fr)

**Abstract.**

The Arctic Ocean is experiencing profound environmental changes due to climate change, with Net Primary Production
(NPP) broadly projected to increase this century. This study analyzes NPP trends and their drivers across pan-Arctic and
sub-regional scales throughout the 21st century, comparing Coupled Model Intercomparison Project Phase 6 (CMIP6) and
Phase 5 (CMIP5) projections to assess how model generations differ. Using a multi-model approach, we assess projections
for different Phytoplankton Functional Types (PFTs), diatoms and nanophytoplankton, and examine the role of physical
and biogeochemical constraints including light, nutrient, and temperature limitations. Our results reveal that Arctic Ocean
NPP increases are primarily driven by reduced sea ice cover, leading to longer ice-free seasons in the expanding seasonal
ice zone. However, NPP changes exhibit pronounced spatial heterogeneity, with strong increases in Arctic inflow shelf
regions, tempered by decreases in Baffin Bay and Nordic Seas. These differences are due to the varying balance between
physical and biogeochemical NPP constraints across the Arctic Ocean. The multi-model mean Arctic Ocean NPP increase
is four times larger in CMIP6 than in CMIP5, under comparable radiative forcing, with a three times higher uncertainty at
the end of the century. This difference is attributed to higher baseline nutrient levels in CMIP6, combined with more
pronounced sea ice loss and greater warming than in CMIP5. Key aspects to better simulate future Arctic Ocean NPP
remain the representation of present-day nutrient levels, light transmission through sea ice and reduced model uncertainty
in climate sensitivity.

## 1 Introduction

Phytoplankton are marine microorganisms that serve as primary producers in ocean ecosystems, forming the foundation of
marine food webs (Vincent & Laybourn-Parry, 2008). Their development depends on biomass accumulation and growth
rates, which are controlled by environmental factors such as light availability, nutrient concentrations and temperature.
Marine Net Primary Production (NPP) represents a fundamental indicator of phytoplankton activity, quantifying net carbon
fixation after losses due to cellular respiration and maintenance processes are accounted for. As an indicator of primary
producer activity, NPP drives the biological ocean carbon pump and therefore plays a key role in long-term ocean carbon
sequestration (Sarmiento, 2013).

This dependence of NPP on environmental conditions makes it sensitive to climate change. Global NPP has declined by
$-2.1\%$ per decade from 1998 to 2015 according to satellite observations (Gregg & Rousseaux, 2019). According to model
hindcast simulations NPP has declined by -6.5% since 1960 (Laufkötter et al., 2015) and is projected to continue to decline
throughout the 21[st] century (Bopp et al., 2013). This decrease is particularly apparent in tropical and mid-latitude regions



where vertical mixing and nutrient supply decline due to enhanced upper-ocean stratification (Doney, 2006). In contrast, at high latitudes, and in particular in the Arctic Ocean where NPP is primarily light limited, observations suggest a recent increase in NPP, rising by 57% between 1998 and 2018 (Lewis et al., 2020).

The Arctic Ocean is particularly sensitive to the effects of climate change (Kwiatkowski et al., 2020; Laufkötter et al., 2015; Vancoppenolle et al., 2013) and is projected to experience warming nearly four times the global average (Rantanen et al., 2022) . This warming results in ocean freshening through increased freshwater input from multiple sources, including melting glaciers and sea ice, enhanced precipitation, and greater river inflow. Moreover, climate change also impacts global ocean currents such as the Atlantic Meridional Overturning Circulation (AMOC), which is an important regulator of the climate system, and which could in turn have an impact on the Arctic Ocean and thus on NPP (Weijer et al., 2020). In polar regions, stratification is primarily driven by salinity gradients rather than temperature differences, making this freshening particularly impactful for ocean mixing and circulation patterns (Timmermans & Marshall, 2020).

The Arctic Ocean phytoplankton community consists of diatoms and smaller taxa including coccolithophores, prymnesiophytes, flagellates, and picoeukaryotes (Ardyna & Arrigo, 2020). Climate change is projected to drive a compositional shift toward smaller phytoplankton dominance (Bopp et al., 2005), fundamentally altering marine food web structure and efficiency. This size-structure transition extends food chain length and reduces trophic transfer efficiency, consequently diminishing carbon pump effectiveness. The shift particularly weakens the biological carbon pump through two mechanisms: smaller phytoplankton are more efficiently recycled within the microbial loop, while larger phytoplankton like diatoms contribute disproportionately to carbon export via gravitational sinking. These structural modifications have cascading effects on both ecosystem functioning and carbon sequestration capacity (Grebmeier et al., 2010; Ward et al., 2012). Furthermore, this phytoplankton community restructuring propagates through upper trophic levels, ultimately impacting commercial fisheries and marine food security (Ardyna & Arrigo, 2020; Hegseth & Sundfjord, 2008; Neukermans et al., 2018). The implications extend beyond primary productivity to encompass Arctic Ocean biogeochemical cycles and the region's role in global climate regulation.

Arctic Ocean NPP has increased by >50% over recent decades, according to satellite observations (Arrigo et al., 2015 ; Lewis et al., 2020). This increase is largely attributed to the decrease in sea ice coverage and thickness, as well as associated changes in sea ice scape , which enhances light penetration into the water column and fosters higher phytoplankton productivity (Lannuzel et al., 2020). Additionally, riverine fluxes and coastal erosion supply a substantial amount of nutrients, supporting approximately one-third of Arctic Ocean productivity (Terhaar et al., 2021). This Arctic Ocean NPP is projected to experience a continued increase in NPP under climate change (Tagliabue et al., 2021). Future sea ice retreat will further extend the open-water season duration, enhance light availability and stimulate phytoplankton growth. However, the melting of sea ice will also introduce large volumes of freshwater, which may further stratify the water column, reducing nutrient transport from deeper waters and potentially modifying NPP dynamics (Pabi et al., 2008; Popova et al., 2010).

CMIP5 Arctic Ocean NPP projections exhibited considerable model divergence with even the sign of future Arctic Ocean NPP anomalies uncertain (Vancoppenolle et al. 2013). However, there is an agreement on the several mechanisms: a decreasing sea ice extent (and thus an increasing light availability) and nitrate concentration, almost reaching the oligotrophy onset in 2100. The projection of temperature wasn't studied as a mechanism influencing NPP (Vancoppenolle et al. 2013).



Despite the recognized importance of environmental drivers in controlling Arctic phytoplankton productivity, several critical knowledge gaps remain. First, it is unclear how projections of key environmental factors (light availability, nutrient concentrations, and temperature) have evolved between CMIP5 and CMIP6 model generations, and whether improved model physics in CMIP6 has led to more consistent or divergent projections. Second, the relative importance of these
drivers in CMIP6 varies spatially across the Arctic Ocean, but systematic analysis of these spatial patterns is lacking. Third, different PFTs respond differently to environmental changes, yet how these differential responses are captured across regions remains poorly understood. Finally, while previous studies have identified sign inconsistencies in CMIP5 NPP projections (Vancoppenolle et al., 2013), the mechanisms underlying the substantially larger NPP increases projected by CMIP6 models have not been systematically analyzed in the Arctic Ocean (Tagliabue et al., 2021). These gaps limit our
ability to assess the reliability of future Arctic Ocean productivity projections and their implications for marine ecosystems and biogeochemical cycles.

To address this knowledge gap, we investigate the environmental drivers of phytoplankton growth using both CMIP6 and CMIP5 models. Our analysis aims to understand how projections and associated uncertainties have changed between the two model generations over the 21st century, examine spatial variations in these projections across the Arctic Ocean, and
assess how different phytoplankton functional types respond to these environmental changes.

## 2 Material and Methods

### 2.1 Model selection

We used output from the Coupled Model Intercomparison Project Phase 6 ((Eyring et al., 2016)) downloaded from the Earth System Grid Federation (ESGF) servers (Table 1). For each model, we prioritized selecting the ensemble member
r1i1p1f1 when available. If r1i1p1f1 was not available, the ensemble member most similar to it, such as r2i1p1f1, r1i1p1f2, or the closest alternative was chosen. The historical period spans 1850-2014, while the SSP scenarios cover 2015-2100. Four SSP scenarios where selected: SSP1-2.6, SSP2-4.5, SSP3-7.0 and SSP5-8.5 (Meinshausen et al., 2020).

To compare with the previous generation of models, from the Coupled Model Intercomparison Project Phase 5 (Taylor et al., 2012), we downloaded output from ESGF servers (Table 2) for only the scenario RCP8.5 (Riahi et al., 2011). As with
CMIP6, we prioritized r1i1f1 when possible. The historical period spans from 1850 to 2005 while the RCP8.5 scenario covers the period 2006-2100.

### 2.2 Models diagnostics

#### 2.2.1 Study region and interpolation methods

The Arctic Ocean was defined as the waters above 66.5°N latitude and divided into 10 different regions, following (Arrigo
& van Dijken, 2011). Both model output and observations were interpolated onto a regular latitude-longitude grid with a resolution of 360×180 using the CDO (Climate Data Operators) software with the remapdis interpolation method.

#### 2.2.2 NPP diagnostics

Model outputs are not always available for all variables of interest, so we categorized the models into two groups based on data availability (Table 1), corresponding to the two parts of our study: In a first part, we examined the prognostic
environmental variables that may influence NPP, namely $NO_3$ concentrations, sea ice concentrations and sea surface temperatures (respectively named in CMIP6: no3, sic and tos) and in the second part we investigated the light, nutrient and



thermal limitation terms that directly impact phytoplankton growth rates. This first part of study used outputs from Group 1 and Group 2 models. The second part of our study focuses on models from Group 2, which provide direct limitation term diagnostics. These limitation terms are derived from the variables mentioned above and represent key factors influencing phytoplankton growth rate for a given phytoplankton type:

$$\mu = \mu_{max} \times T_f \times L_{lim} \times N_{lim} \tag{1}$$

where $\mu_{max}$ is the maximum growth rate, $L_{lim}$ and $N_{lim}$ are the limitation terms for the light and nutrient respectively, and $T_f$ represents the temperature function.

### 2.2.3 Nutrients, SST, Sea ice and limitation factors

Nitrate ($NO_3$) was selected as the limiting nutrient, as the Arctic Ocean is primarily nitrogen-limited, and ammonium ($NH_4$) is not available for all models. The final diagnostic used for analysis was the vertically-averaged concentration over the

upper 100 m. An oligotrophy threshold was set at 1.6 mmol/m$^3$, i.e., at the half-saturation constant of diatoms for $NO_3$ uptake (Vancoppenolle et al., 2013).

As a diagnostic of ice coverage, we used monthly mean sea ice concentration (SIC) outputs. From the latter, we derived the September sea ice area (SSIA), the mean September sea ice concentration (SSIC), and the mean ice-free season duration (IFSD). The IFSD was defined as the number of months with sea ice concentration lower than 15% (Lebrun et al., 2019).

We analyzed growth rates for the two PFTs available in these models: diatoms and miscellaneous phytoplankton. The direct limitation terms provided by the models include: the light limitation term for miscellaneous phytoplankton ($L_{lim}^{misc}$, named limirrmisc in CMIP6 terminology), the light limitation term for diatoms ($L_{lim}^{diat}$, limirrdiat), the nutrient limitation term for miscellaneous phytoplankton ($N_{lim}^{misc}$, limnmisc) and the nutrient limitation term for diatoms ($N_{lim}^{diat}$, limndiat). These limitation terms can vary between 0 and 1, with lower values indicative of greater constraint on phytoplankton growth

rates.

The temperature function ($T_f$) is not directly provided by the models, but it is calculated using the Eppley function ($K_{Eppley}$ = 0.063°C$^{-1}$) except for UKESM1-0-LL which calculates the temperature function differently, as $T_f = 1.006^T$, based on vertically gridded temperature (3Dtemp, thetao in CMIP6 terminology). Monthly $T_f$ was computed for each depth level and weighted according to the biomass of each PFT (phymisc or phydiat in CMIP6 terminology) to obtain PFT-specific

depth integrated temperature functions for each model ($T_f^{misc}$, and $T_f^{diat}$).



| | Model (reference) | Ocean - Sea-ice - Biogeochemical components | Available Variables | historical | SSP1-2.6 | SSP2-4.5 | SSP3-7.0 | SSP5-3.4 | SSP5-8.5 |
|---|---|---|---|---|---|---|---|---|---|
| **Group 1** | **CESM2** (Danabasoglu et al., 2020) | POP2 - CICE5 - MARBL-BEC | NPP, NO3, SIC, SST | x | x | x | x | | x |
| | **CESM2-WACCM** (Danabasoglu, 2019) | POP2 - CICE5 - MARBL-BEC | NPP, SIC, SST | x | x | x | x | x | x |
| | | | NO3 | x | x | x | x | | x |
| | **CanESM5** (Swart et al, 2019) | NEMO3.4 - LIM2 - CMOC | NPP, NO3 | x | x | x | x | | x |
| | | | SIC, SST | x | x | x | x | x | x |
| | **CanESM5-CanOE** (Christian et al, 2021) | NEMO3.4 - LIM2 - CanOE | NPP, NO3, SIC, SST | x | x | x | x | | x |
| | **MIROC-ES2L** (Hajima et al 2020) | COCO - OECO2 | NPP, NO3, SIC, SST | x | x | x | x | x | x |
| | **MPI-ESM1-2-HR** (Mauritsen et al., 2019) | MPIOM - HAMOCC6 | NPP, NO3, SIC, SST | x | x | x | x | | x |
| | **MPI-ESM1-2-LR** (Mauritsen et al., 2019) | | NPP, NO3, SIC, SST | x | x | x | x | | x |
| | **MRI-ESM2-0** (Yukimoto et al, 2019a) | MRICOM4 - NPZD | NPP, NO3, SIC, SST | x | | | | | x |
| | **NorESM2-LM** (Tjiputra et al 2020) | BLOM - CICE5 - iHAMMOC | NPP, SIC, SST | x | x | x | x | x | x |
| | | | NO3 | x | x | | | | x |
| **Group 2** | **CNRM-ESM2-1** (Séférian et al, 2019) | NEMOv3.6 - GELATOv6 - PISCESv2-gas | NPP, SIC, SST | x | x | x | x | x | x |
| | | | $L_{lim}^{diat}, L_{lim}^{misc}, N_{lim}^{diat}, N_{lim}^{misc}$, phydiat, phymisc, 3Dtemp | x | | | | | x |
| | | | NO3 | x | x | x | x | | x |
| | **GFDL-ESM4** (Dunne et al 2020b ; Stock et al, 2020) | MOM6 - SIS2 - COBALTv2 | NPP, NO3, SIC, SST | x | x | x | x | | x |
| | | | $L_{lim}^{diat}, L_{lim}^{misc}, N_{lim}^{diat}, N_{lim}^{misc}$, phydiat, phymisc, 3Dtemp | x | | | | | x |
| | **IPSL-CM6A-LR** (Boucher et al, 2020) | NEMOv3.6 - LIM3 - PISCESv2 | NPP, NO3, SIC, SST | x | x | x | x | x | x |
| | | | $L_{lim}^{diat}, L_{lim}^{misc}, N_{lim}^{diat}, N_{lim}^{misc}$, phydiat, phymisc, 3Dtemp | x | | | | | x |
| | **UKESM1-0-LL** (Sellar et al, 2019) | NEMO v3.6 - CICE - MEDUSA-2 | NPP, SIC, SST | x | x | x | x | x | x |
| | | | $L_{lim}^{diat}, L_{lim}^{misc}, N_{lim}^{diat}, N_{lim}^{misc}$, phydiat, phymisc, 3Dtemp | x | | | | | x |
| | | | NO3 | x | x | x | x | | x |

*Available Simulations* (spanning header for the scenario columns)

**Table 1:** *Description of CMIP6 models used in this study, along with the associated available variables and scenario simulations. The first group includes models used to project environmental conditions (NPP, NO₃, SIC, and SST). The second group includes models used for both environmental projections and analysis of phytoplankton growth limitation terms (Nlim, Llim and Tf).*



| Model (reference) | Ocean - Sea-ice -biogeochemical components | Available Variables | Available Simulations |
|---|---|---|---|
| **CESM1-BGC** (Moore et al., 2013) | POP - CICE4 – BEC | SST | historical, RCP8.5 |
| **CMCC-CESM** (Fogli & Iovino, 2014) | NEMO3.4 - CICE4 – CMCC | NO3, SST | historical, RCP8.5 |
| **GFDL-ESM2G** (Dunne et al., 2013) | ESM2G – MOM4p1 – TOPAZ2 | NPP, NO3, SIC, SST | historical, RCP8.5 |
| **GFDL-ESM2M** (Dunne et al., 2013) | ESM2M – MOM4p1 – TOPAZ2 | NPP, NO3, SIC, SST | historical, RCP8.5 |
| **HadGEM2-ES** (The HadGEM2 Development Team: G. M. Martin et al., 2011) | HadGEM2 – Diat-HadOCC | SST | historical, RCP8.5 |
| **IPSL-CM5A-LR** (Dufresne et al., 2013) | NEMO - LIM2 - PISCES | NPP, NO3, SIC, SST | historical, RCP8.5 |
| **IPSL-CM5A-MR** (Dufresne et al., 2013) | NEMO - LIM2 - PISCES | NPP, SIC, SST | historical, RCP8.5 |
| **MIROC-ESM** (Watanabe et al., 2011) | COCO – NPZD-type | NPP, SIC | historical, RCP8.5 |
| **MIROC-ESM-CHEM** (Watanabe et al., 2011) | COCO – NPZD-type | NPP, SIC | historical, RCP8.5 |
| **MPI-ESM-LR** (Ilyina et al., 2013) | – HOMOCC5.2 | NPP, NO3, SIC, SST | historical, RCP8.5 |
| **MPI-ESM-MR** (Ilyina et al., 2013) | – HOMOCC5.2 | NPP, SIC, SST | historical, RCP8.5 |
| **NorESM1-ME** (Bentsen et al., 2013) | MICOM – CICE4 – HAMOCC-ME | NO3, SST | historical, RCP8.5 |

*Table 2: CMIP5 models used in this study, along with the available variables and scenario simulations.*

**2.3 Data used for model evaluation**

5   The model evaluation datasets include observational products of primary production, nitrate concentration, sea ice concentration, and SST.

*Primary Production*

The primary production dataset used in this study provides recalculated depth-integrated NPP values based on satellite-derived chlorophyll concentrations, sea surface temperature, and sea ice cover (Lewis et al., 2020). This dataset spans 10   1998–2018 and offers satellite-based estimates in the Arctic Ocean. It was regridded onto a 1° × 1° spatial resolution. Only NPP data from March 1st to September 30th were considered for model evaluation, due to the lack of ocean color observations during the dark season.

*Nitrate (NO₃)*

To evaluate nitrate concentrations we used a climatology derived from in situ hydrographic observations from the World 15   Ocean Atlas 2018 (Garcia et al., 2019). This dataset provides an average nitrate concentration from 1978 to 2017 on a 1° × 1° grid. We selected nitrate concentrations for the upper 100 m of the water column, as this depth range is most relevant





to phytoplankton growth. However, nitrate data in the Arctic Ocean remain sparse, particularly in summer, and may be subject to significant biases (Popova et al., 2012).

*Sea ice concentration*

Passive microwave satellite sea ice concentration fields were taken from the Global sea ice concentration data record
v3 (OSI SAF, 2022). This dataset provides daily averaged sea ice cover (in percentage) at a 25 km spatial resolution, covering the period 1978–2022.

*Sea Surface Temperature (SST)*

The sea surface temperature dataset is derived from the Climate Change Initiative (CCI) satellite product, covering the period 1982–2019 (Embury., 2024). This dataset provides a long-term, high-quality observational record of SST, allowing
for comparison with model outputs.

## 3 Results

### 3.1 Model evaluation

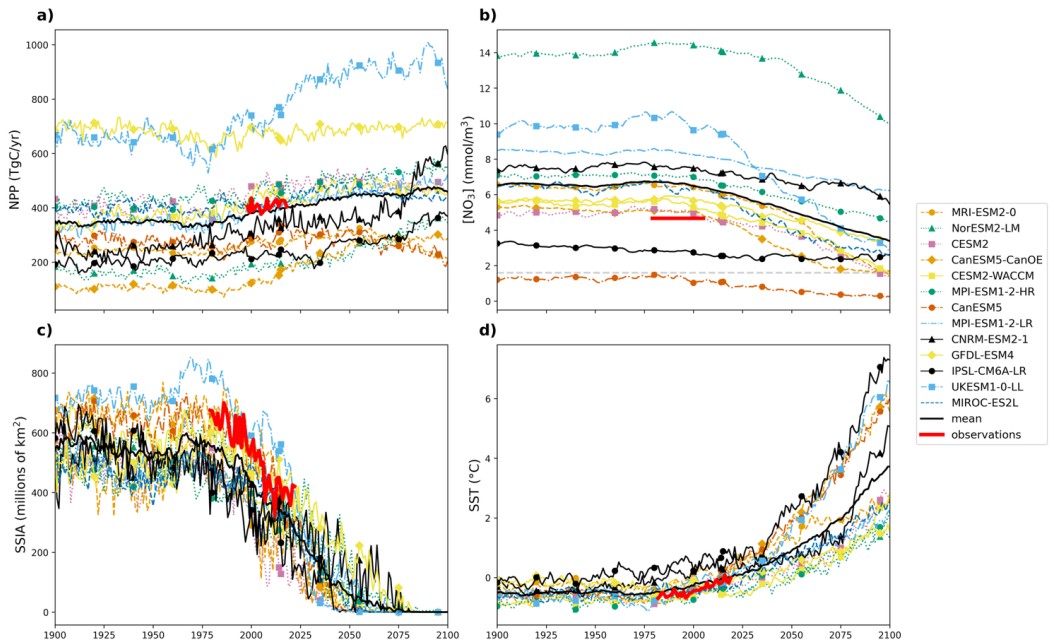

**Figure 1.** *Projections of Arctic Ocean integrated NPP (a), mean upper 100m NO₃ (b), September Sea Ice Area (c) and mean SST (d)*
*from individual CMIP6 models (group 1), using historical and SSP5-8.5 simulations from 1900 to 2100. Each color represents a model, the black line is the multi-model mean, and the horizontal grey dashed line represents NO₃ concentration consistent with oligotrophy.*



| Variable | Period | Observed mean | CMIP6 mean | Observed trend (/yr) | CMIP6 trend (/yr) |
|---|---|---|---|---|---|
| NPP (TgC/yr) | 1998-2018 | 308 | 389 | 6.73 | 1.30 |
| SSIA (10⁶ km²) | 1979-2022 | 5.28 | 4,31 | -0.059 | -0.056 |
| SST (°C) | 1982-2019 | -0,402 | -0,247 | 0.019 | 0.013 |

***Table 3:*** *Observed and simulated NPP, SSIA and SST values throughout the observational period. Nitrate concentrations are not given as an Arctic Ocean time series is not available.*

Simulated ranges of NPP, upper-100m $NO_3$, SSIA and SST encompass data-based estimates over the observational period (Fig. 1). While the simulated trends in Arctic Ocean warming and declining SSIA are broadly consistent with observations,
5    simulated increases in NPP are low biased. The multi-model mean NPP is slightly higher than observed, over 1998-2018 period over which all models consistently simulate an NPP increase.  Observed NPP increases from 256 Tg C/yr in 1998 to 391 Tg C/yr in 2018, increasing by 135 Tg C/yr in 21 years. The increase in NPP in models is lower, increasing from 354 Tg C/yr to 394 Tg C/yr in 21 years (Fig. 1a, Table 3). The multi-model mean nutrient concentration for the period 1979-2005 is 6.48 mmol/m³, which is higher than the observations of 4.67 mmol/m³ for the same period. However, this
10   discrepancy is not unexpected, as Arctic Ocean $NO_3$ observations contain significant uncertainties due to limited accessibility beneath sea ice cover (Fig. 1b). Models and observations agree on a loss of sea ice coverage, although the modeled rate of decrease is lower than observed (Fig. 1c), with sea ice retreat proceeding faster in observations than simulated (Table 3), confirming previous studies (Notz & Community, 2020). Similarly, observed Arctic Ocean warming is ~50% higher than that simulated over the same period (Fig. 1d, Table 3).



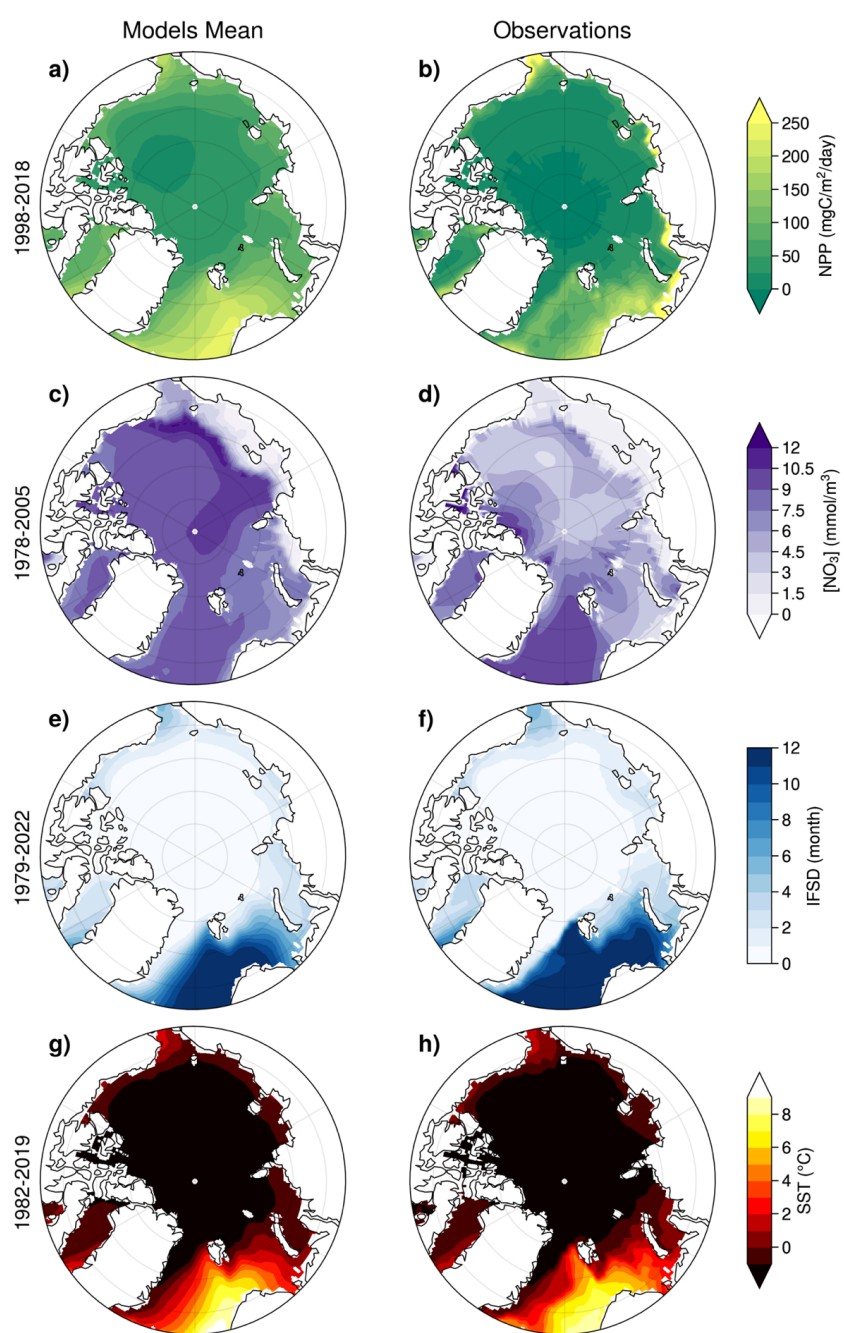

**Figure 2.** *NPP, NO$_3$, IFSD, and SST, over the recent past, according to the multi-model mean (left) and observational references (right). The averaging period corresponds to the availability of observations prior to 2015. Model outputs are combined from historical simulations for years prior to 2015 and SSP5-8.5 thereafter. For NPP, the March to September average is shown.*




The spatial distribution of key variables across the Arctic Ocean is generally well represented by the multi-model mean and comparable to observational references despite some regional discrepancies that reflect both model limitations and observational uncertainties. March-September NPP resembles satellite-derived estimates, with the highest values in the Nordic Seas, the Bering Strait, and along the Siberian shelf, while the Arctic Basin exhibits the lowest NPP values (Fig. 2a

and b). The multi-model mean IFSD and SST spatial distributions also closely aligned with observations. The warmest regions (Nordic Seas, Barents Sea, and Chukchi Sea) correspond to areas with the least sea ice (Fig. e, f, g and h). While the multi-model mean represents low values coastal $NO_3$ concentrations, in the Chukchi Sea and along the eastern Arctic Basin. It may overestimate nitrate concentrations in the Arctic Basin, though observations in this region remain limited. The coasts of Kara, Laptev and East Siberian Seas show low $NO_3$ values — of less than 1.5 mmol/m³ in both models and

observations (Fig. 2c and d).

## 3.2 Projections of NPP and its environmental drivers

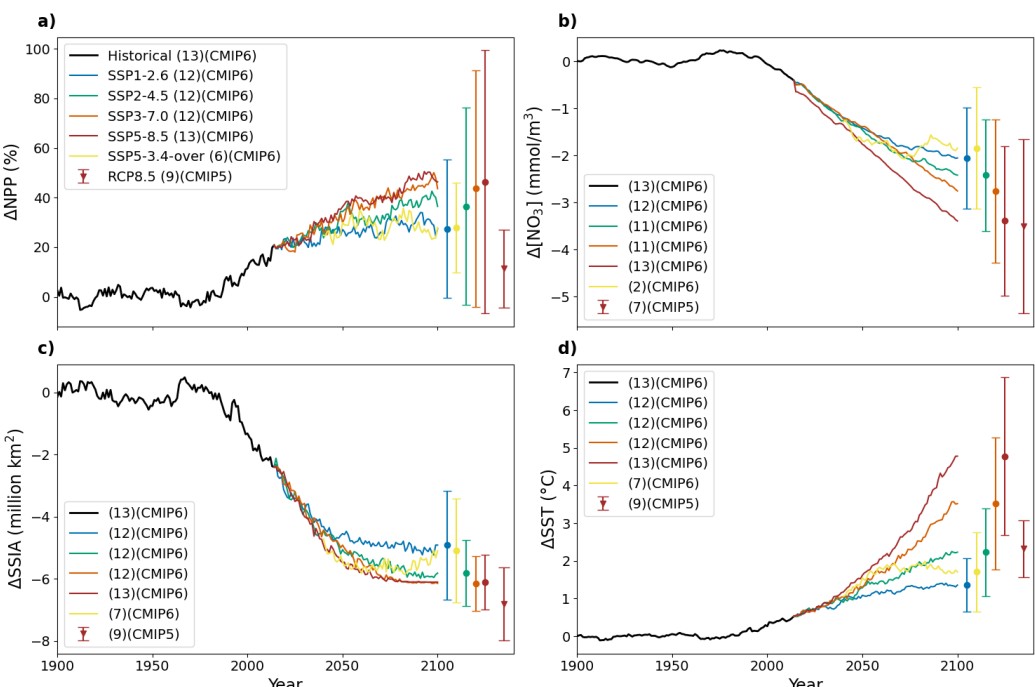

**Figure 3.** *CMIP6 projected Arctic Ocean anomalies of NPP (a), NO₃ (b), SSIA (c) and SST (d) over the period 1900-2100 relative to*
*1850-1899, in the different emission scenarios. Vertical bars represent multi-model means and standard deviation in 2100. The last vertical bar and triangle represents the value for RCP8.5 of CMIP5 in 2100. The numbers within parentheses give the number of available models.*

Across all SSPs, Arctic Ocean NPP and SST are projected to increase this century, while nitrate concentration and SSIA decline (Fig. 3). However, the magnitude of change of each variable depends on the scenario, with generally greater

anomalies and associated uncertainty under greater radiative forcing. An exception to this, is SSP5-3.4, for which there is lower model uncertainty, possibly due to the small model ensemble size (Fig. 3).



Projected Arctic Ocean NPP and nitrate inter-model uncertainty is greater than inter-scenario uncertainty by the end of the century (Fig. 3a and b). Contrary to the other variables, SSIA exhibits reduced uncertainty under higher radiative forcing scenarios. This likely reflects the strong model agreement that the Arctic Ocean will be ice-free by the end of the century (Fig. 3c). Surface ocean warming shows significant sensitivity across scenarios, with the greatest warming (multi-model

mean +4.78°C in 2081-2100), simulated in SSP5-8.5 (Fig. 3d).

Projected twenty-first century NPP increases are highly divergent between CMIP6 and CMIP5 scenarios of comparable high radiative forcing (Fig 3a). For RCP8.5, the relative anomaly in NPP reaches $11.5 \pm 15.7$ %. In contrast, SSP5-8.5 projects a much larger increase of $46.4 \pm 53.7$ %. This highlights greater CMIP6 model agreement of increasing future NPP alongside a greater than 3-fold increase in associated uncertainty (Fig 3a). The projected twenty-first century Arctic Ocean

$NO_3$ decline is highly similar between SSP5-8.5 and RCP8.5 (respectively $-3.38 \pm 1.58$ mmol/m$^3$ and $-3.50 \pm 1.85$ mmol/m$^3$, Fig. 3b). September sea ice area loss is lower in SSP5-8.5 than in RCP-8.5, likely due to larger lower sea ice extent during the period 1881-1900 (Fig. 3c), while surface ocean warming is two times higher in SSP5-8.5 ($4.78 \pm 2.09$ °C) than RCP8.5 ($2.32 \pm 0.75$ °C) with higher associated uncertainty (Fig. 3d). With respect to the emergence of oligotrophic conditions, CanESM5 stands out by simulating oligotrophic $NO_3$ levels throughout the historical period and

future simulations. In comparison, CanESM5-CanOE, CESM2-WACCM, CESM2, and MRI-ESM2-0 cross this threshold in 2083, 2093, 2087, and 2092 of SSP5-8.5 respectively, while all other models maintain $NO_3$ concentrations above this threshold throughout simulations (Fig. 1b).



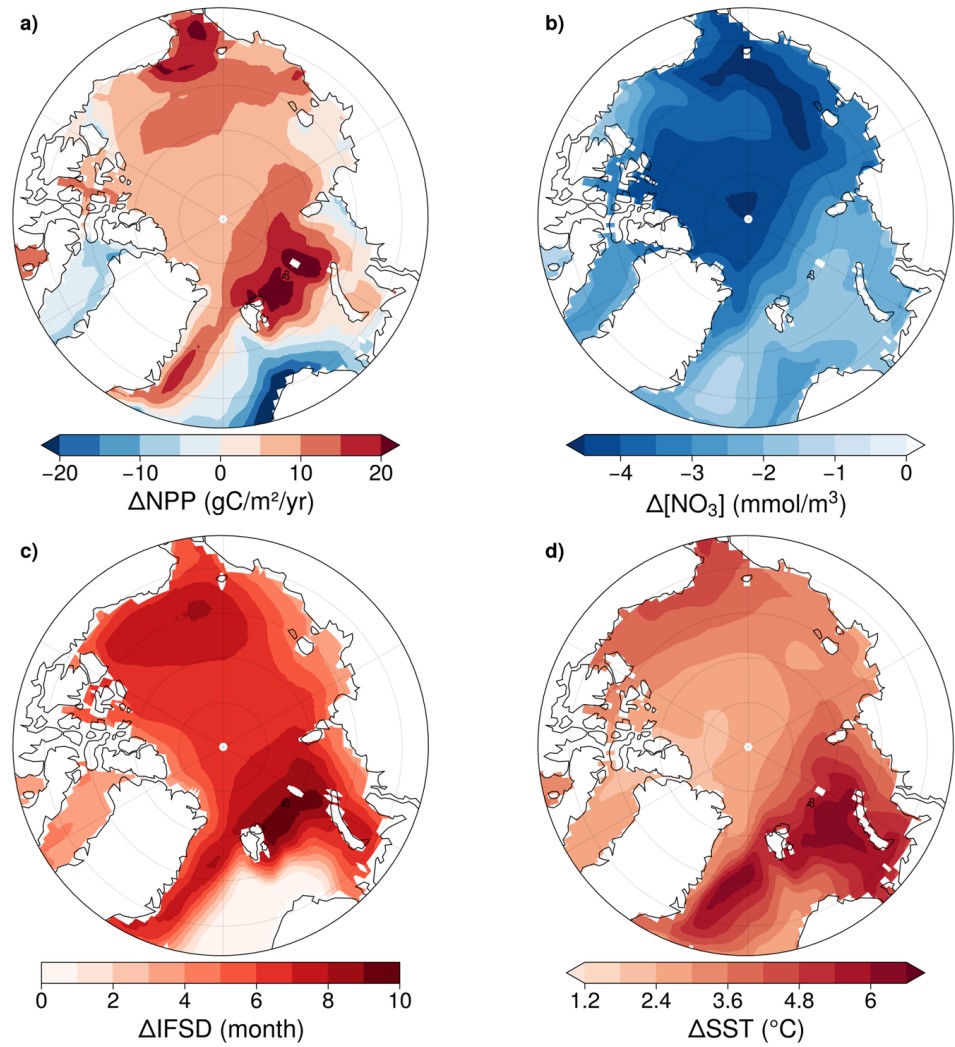

**Figure 4.** *CMIP6 multi-model mean anomalies of NPP (a), NO₃ (b), IFSD (c) and SST (d). Anomalies are in 2081-2100 of SSP5-8.5 relative to 1995-2014 of the historical simulation.*

Multi-model mean declines in NO$_3$ and increases in IFSD and SST occur across the Arctic Ocean domain, however NPP
exhibits both regional-scale increases and decreases (Fig. 4). NPP increases are greatest in the northern Barents Sea, along
the east coast of Greenland, and in the Chukchi Sea, reaching more than 20 gC/m²/yr. In contrast, NPP decreases along the
coast of the Kara Sea, in the North Atlantic, and in Baffin Bay (Fig. 4a). NO$_3$ concentration decreases across the entire
basin, with the most pronounced reductions in of the Siberian shelf and off the East Siberian Shelf, by more than 4 mmol/m³
(Fig. 4b). The region experiencing the highest increase in the ice-free season duration is the northern Barents Sea, east of
Svalbard Archipelago. In the central basin, sea ice loss is also strong, losing up to more than 6 months of annual ice cover
(Fig. 4c). Barents and Greenland Seas are the regions most affected by rising SST, with the SST increasing by up to 6°C in
these zones (Fig. 4d).




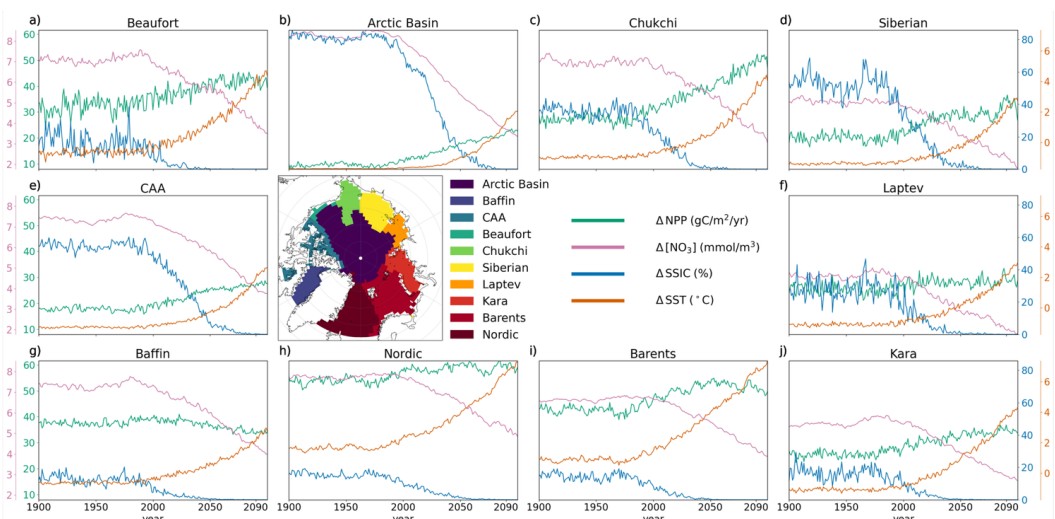

**Figure 5.** *CMIP6 multi-model mean basin-scale evolution of NPP (green), NO₃ (pink), SSIC (blue) and SST (orange), over the period 1900-2100 of the historical and SSP5-8.5 simulations.*

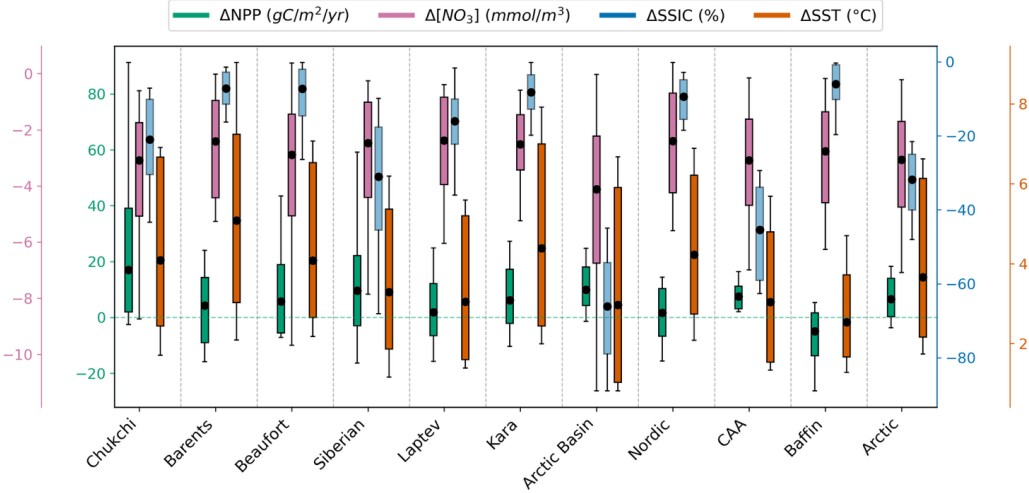

5     **Figure 6**. *CMIP6 ensemble anomalies of NPP (green), NO₃ (pink), SSIC (blue) and SST (orange) for each Arctic basin and the entire Arctic Ocean domain. Each boxplot represents the multi-model mean, interquartile range and ensemble outliers.*

Across all Arctic sub-regions, multi-model mean NPP, NO₃, SSIC and SST are relatively stable until ~1980, after which all basins begin to exhibit varying levels of surface ocean warming alongside NO₃ and SSIC declines (Fig. 5). The sign of regional-scale anomalies in NO₃, SSIC and SST is consistent across the CMIP6 ensemble however model uncertainty varies

10   across regions (Fig. 5 and 6).

The area-specific Arctic Ocean ΔNPP is relatively low compared to other regions, with an ensemble range of –3.62 gC/m²/yr to 18.3 gC/m²/yr for the entire Arctic Ocean and –2.52 to 91.3 gC/m²/yr for the Chukchi Sea, where the increase and associated uncertainty is the highest. The decrease in NO₃ concentration is relatively consistent across Arctic sub-





regions, ranging from approximately –2.38 in Laptev Sea to –4.12 mmol/m$^3$ in the Arctic Basin. Temperature is rising across all basins, with an average increase of approximately 3.66 °C across the whole Arctic. The highest annual temperature uncertainty is found in the Barents Sea, with a model spread of 6.95°C, while the lowest uncertainty occurs in Baffin Bay, with a spread of 3.42°C (Fig. 6).

5 This region, Baffin Bay differs from other regions as the only area where multi-model mean NPP is projected to decline under SSP5-8.5. During the historical period (1995-2014), this basin already has low September sea ice concentration, meaning that even though it becomes ice-free by the end of the century under SSP5-8.5, the anomaly remains small. Baffin Bay is projected to experience the lowest multi-model mean surface ocean warming (2.53 °C) compared to other Arctic ocean basins, with a low uncertainty (Fig. 6).



### 3.3 Phytoplankton growth limitation terms

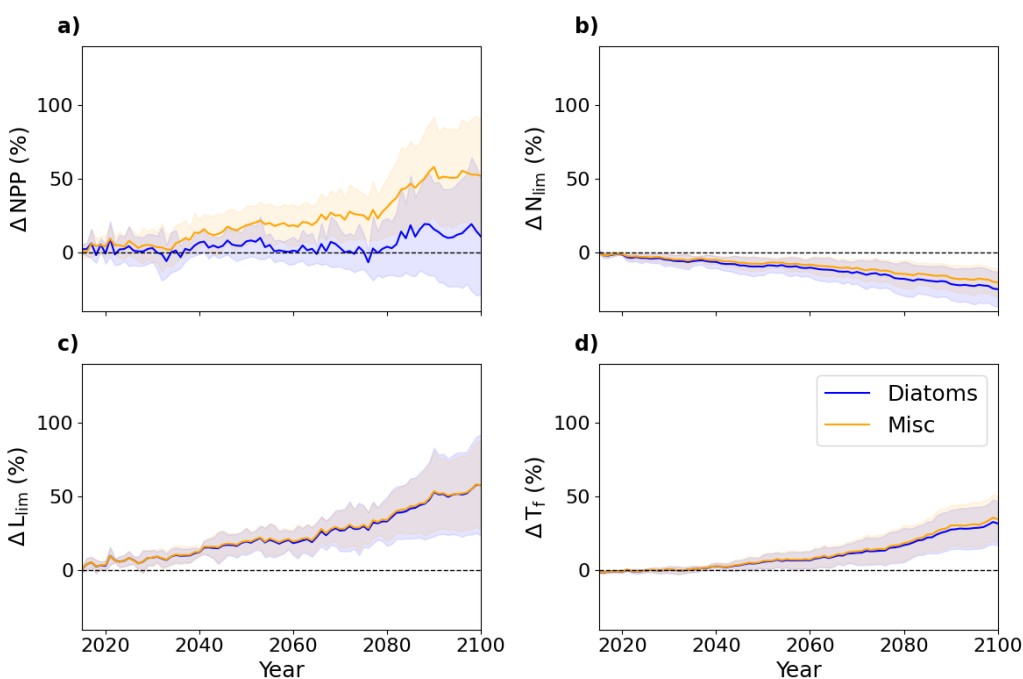

***Figure 7.*** *Multi-model mean SSP5-8.5 Arctic Ocean anomalies of diatom (blue) and miscellaneous phytoplankton (orange) NPP (a), nutrient limitation (b), light limitation (c) and temperature function (d). Anomalies are relative to 1995-2014 values and multi-model uncertainty, represented as the standard deviation are shaded.*

As phytoplankton growth is the product of multiple limiting factors (Equation 1), the analysis of their relative anomalies throughout the 21$^{st}$ century is necessary to assess their respective contributions to phytoplankton growth rate perturbations and the projected increase in Arctic Ocean NPP. Diatoms exhibit a smaller increase in NPP compared to miscellaneous phytoplankton, with respective increases of approximately 11 % and 52 % relative to 1995-2014. However, the uncertainty associated with both groups is high and increases over SSP5-8.5 (Fig. 7a). The greatest contribution to relative increases in multi-model mean phytoplankton growth rates is relaxed light limitation which is augmented by the influence of warming on phytoplankton temperature functions and slightly offset by enhanced nutrient limitation (Fig 7). The $L_{lim}$ and $T_f$ increase is consistent between PFTs (Fig. 7b,c). However, $\Delta L_{lim}$ uncertainty increases sharply, exceeding ±30% by 2100 for both PFTs while $\Delta T_f$ and $\Delta N_{lim}$ uncertainty is between ±10% and ±20% by 2100 (Fig. 7b, c and d). Both PFTs are increasingly nutrient limited in the Arctic Ocean over the 21$^{st}$ century, but diatoms are more impacted by nutrient concentration declines (Fig. 7b).



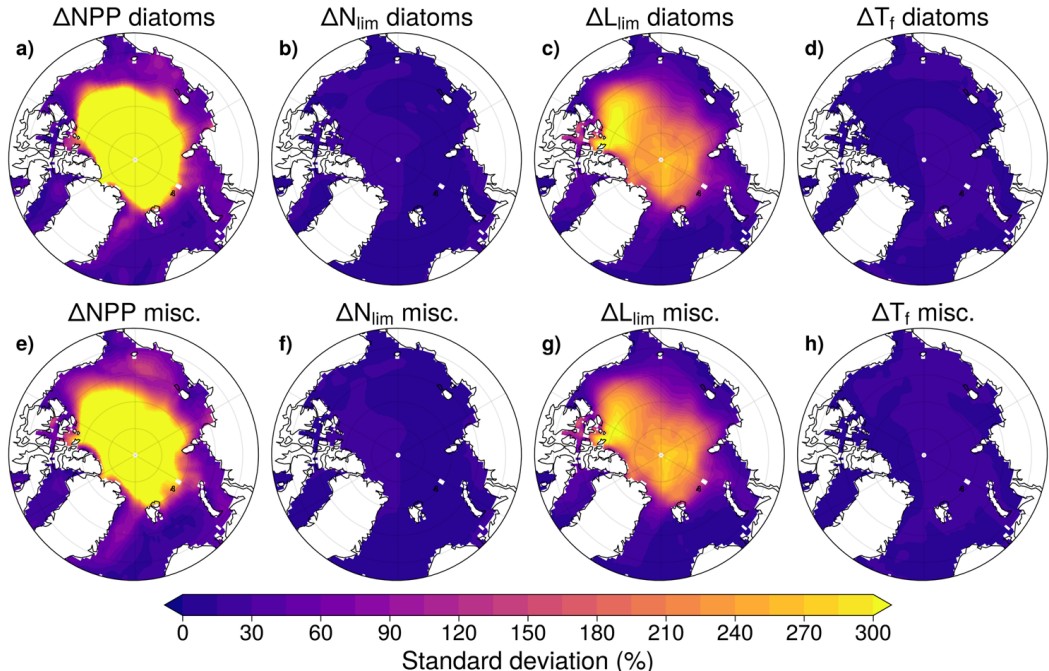

**Figure 8.** *Regional differences in the multi-model standard deviation of anomalies in NPP (a, e), nutrient limitation (b, f), light limitation (c, g) and temperature function (d, h) in 2081-2100 of SSP5-8.5 relative to 1995-2014 of the historical. The upper row represents diatoms and the lower row miscellaneous phytoplankton.*

5    Diatoms and miscellaneous phytoplankton show comparable Arctic Ocean patterns of multi-model uncertainty associated with SSP5-8.5 relative anomalies in NPP and phytoplankton growth limitation terms (Fig. 8). Relative $\Delta$NPP uncertainty is particularly pronounced in the central Arctic Ocean (Fig. 8a and e). This is coincident with the location of greatest $\Delta L_{lim}$ uncertainty which reaches >210 % over the century in this region (Fig. 8c and g). In contrast, $\Delta T_f$ and $\Delta N_{lim}$ exhibit much lower uncertainty, remaining below 30 % across the entire Arctic Ocean (Fig. 8b, d, f and h). Remarkably, the regions

10    of greatest relative NPP uncertainty (central Arctic Ocean) differ from those exhibiting the largest absolute NPP increases (shelf regions, Fig. 4a), highlighting that uncertainty patterns do not necessarily coincide with the magnitude of projected changes.



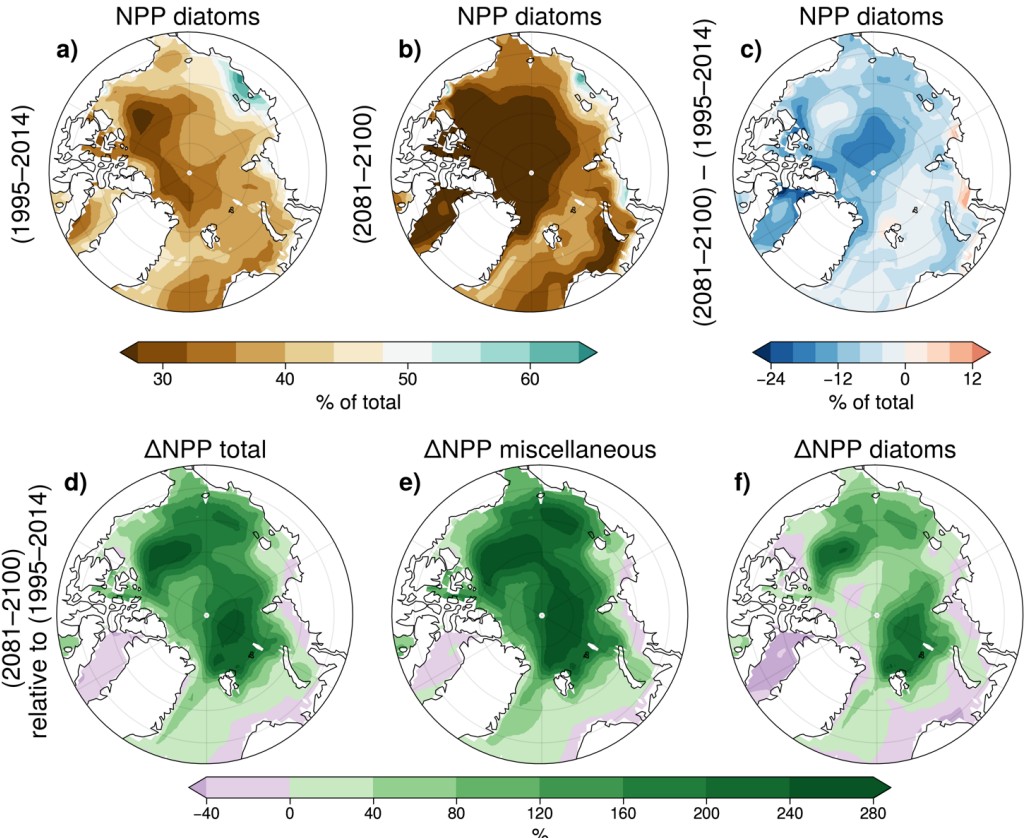

***Figure 9.*** *Multi-model mean fraction of total NPP realized by diatoms during the period 1995-2014 of the historical (a) and 2081-2100 of SSP5-8.5 (b) and the difference between those periods (c). Anomaly of the total NPP in 2081-2100 (d), for miscellaneous phytoplankton € and for diatoms (f) relative to 1995-2014.*

In the two decades 1995-2014, miscellaneous phytoplankton are the dominant contributor to total NPP across the majority of the Arctic Ocean with the exception of the coastal East Siberian Sea where diatom NPP is particularly important (Fig. 9a). Under SSP5-8.5, miscellaneous phytoplankton dominance is enhanced, notably in the central Arctic Ocean where they contribute to up to 70 % of total NPP. In contrast, diatom NPP remains dominant in the coastal East Siberian Sea and

10 exhibits increasing dominance in the Kara and Laptev Seas (Fig. 9b). Relative increases in total NPP are greatest in the central Arctic basin, with >200 % enhancement compared to 1995-2014 values. This spatial pattern is consistent with the distribution of NPP increases associated with each PFT, although increases are higher for miscellaneous phytoplankton than diatoms (Fig. 9e,f). Contrary to this, the NPP of both PFTs, and in particular diatom NPP, declines in the Baffin Bay (-40 % diatom NPP), as well as in parts of the Nordic Seas and on the West Siberian shelf.



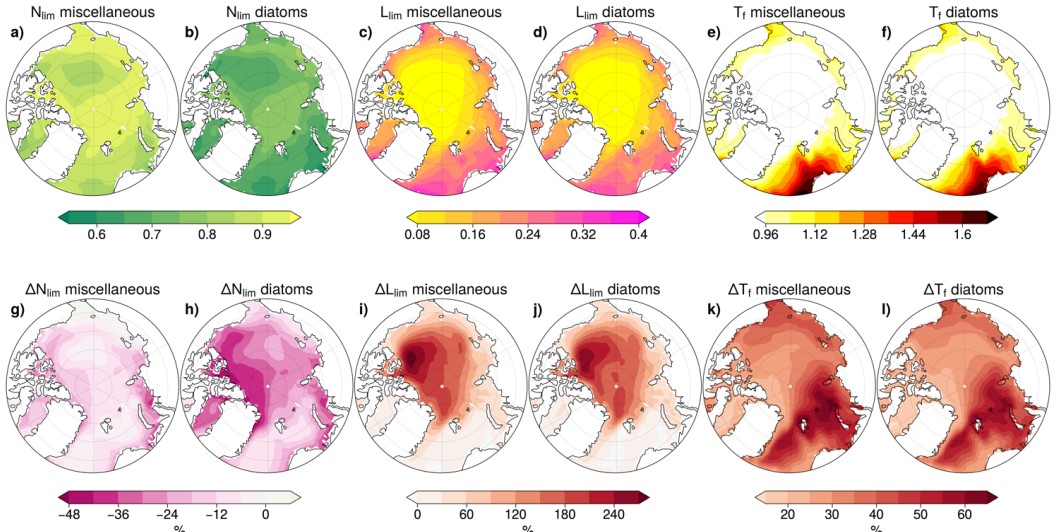

**Figure 10.** *CMIP6 multi-model mean state nutrient limitation, light limitation and temperature function for each PFT in 1995-2014 of the historical simulation (top) and anomalies in 2081-2100 of SSP(-8.5 (bottom).*

In the reference period (1995-2014), $N_{lim}$ is lower for diatoms than for miscellaneous phytoplankton (Fig. Figure a and b). This is indicative of diatoms being less competitive for nutrients across the Arctic Ocean. Furthermore, diatoms are projected to experience stronger declines in $N_{lim}$ compared to miscellaneous phytoplankton under SSP5-8.5 (Fig. 10g and h). In the Canadian Basin and in Baffin Bay, diatom $N_{lim}$ decreases by over 40 %, indicating that diatoms face strongest nutrient constraints there. For miscellaneous phytoplankton, nutrient limitation is also enhanced but to a lesser extent, with

a maximum $N_{lim}$ decrease of less than 20 % (Fig. 10g and h).

For both PFTs, $L_{lim}$ and $T_f$ exhibit similar regional distributions during the reference period (1995-2014). Light limitation is highest in the Nordic and Barents Seas, reaching 0.4, while temperature factors also peak in these regions at values exceeding 1.8. Conversely, both $L_{lim}$ and $T_f$ show their lowest values in the Arctic Basin (Fig. 10c, d, e, and f). The evolutionary patterns of these terms remain consistent between PFTs, though their spatial distributions of change differ.

Light limitation factor increases most substantially in the central Arctic, with an increase of approximately 180% relative to the reference period (Fig 10i and j) showing a strong alleviation of light limitation, while temperature factor increases are most pronounced in the Kara Sea, Barents Sea, and Nordic Seas, exceeding 50% relative to the reference period (Fig 10k and l).





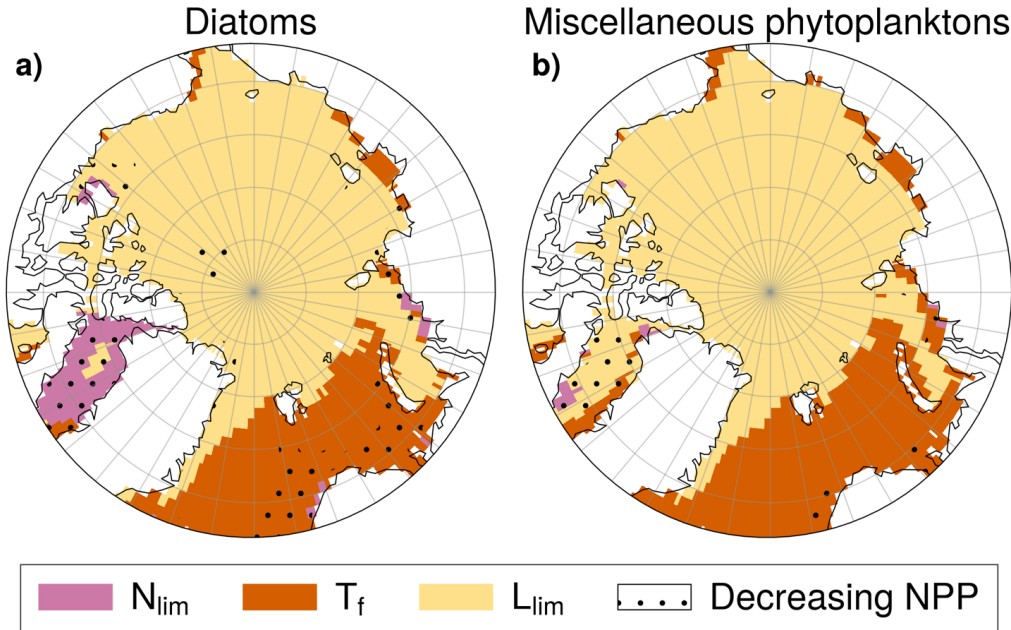

***Figure 11***. *Dominant driver of phytoplankton growth rate changes (Δμ) in SSP5-8.5 for diatoms (a) and miscellaneous phytoplankton (b). Area where* $L_{lim}$ *(yellow),* $T_f$ *(orange) or* $N_{lim}$ *(pink) are the largest contributor to Δμ. NPP anomalies are negative in stippled areas and positive elsewhere.*

In SSP5-8.5 simulations, increasing $L_{lim}$ (reduced light limitation) is the principal driver of multi-model mean increases in phytoplankton growth rates for both diatoms and miscellaneous phytoplankton across the Arctic Ocean, particularly in the central Arctic where this is coincident with NPP increases (Fig. 11). In the Nordic, Barents, and Kara Seas, phytoplankton growth rate changes of both PFTs are mainly driven by increasing $T_f$ (thermal enhancement) and generally consistent with NPP increases, except off the Norwegian coast. In contrast, in Baffin Bay, declining $N_{lim}$ (greater nutrient limitation) is the

dominant contributor to growth rate changes for diatoms and is consistent with declining diatom NPP. For miscellaneous phytoplankton however, increasing $L_{lim}$ is the dominant contributor to growth rate changes. This is indicative of enhanced growth rates but miscellaneous phytoplankton NPP is projected to decline in Baffin Bay (Fig. 11b). This suggests that other mechanisms, such as enhanced zooplankton grazing rates (Rohr et al., 2023) are contributing to projected NPP declines in this region.

**4. Discussion**

Projected twenty-first century increases in Arctic Ocean NPP are higher and more variable in CMIP6 than in CMIP5. The multi-model mean NPP increase is four times larger under comparable high radiative forcing in CMIP6 than in CMIP5 (respectively 46.4 % and 11.5 % compared to the historical period, see Fig. 3a), with uncertainty at the end of the century three times higher in CMIP6 than CMIP5, consistent with previous studies (Tagliabue et al. 2021). The CMIP6 and CMIP5

ensembles also exhibit different temporal evolution of NPP over the course of the 21$^{st}$ century. All CMIP6 models exhibit monotonic increases in Arctic Ocean NPP, whereas CMIP5 models exhibit more diverse responses, with some models showing an initial increase in NPP followed by a subsequent decrease due to the emergence of oligotrophic conditions





(Vancoppenolle et al. 2013). In absolute terms, the most substantial NPP increases in CMIP6 are observed in the inflow shelf regions, that experience the greatest warming, sustained nutrient levels and limited change in light availability. This is consistent with the trend in the current observations (Lewis, van Dijken, et Arrigo 2020). The model uncertainty associated with CMIP6 Arctic Ocean NPP projections is higher than scenario uncertainty (Fig. 3), consistent with global NPP projections and indicative of the finely balanced limitations on NPP, which often compensate one another (Bopp et al., 2013; Kwiatkowski et al., 2020).

The direct influence and balance between climate-driven drivers of phytoplankton growth rates, namely temperature, light and nutrient availability, shifts substantially in CMIP6 compared to CMIP5. The stronger Arctic Ocean surface warming projected in CMIP6 and greater associated uncertainty (Fig. 3), relates to higher and more variable climate sensitivity (Zelinka et al., 2020) with similar Arctic amplification simulated across both ensembles (Hahn et al., 2021). This results in greater thermally-driven increases in phytoplankton growth rates than previously simulated in CMIP5 (Laufkötter et al., 2015; Nakamura & Oka, 2019). The more pronounced Arctic Ocean sea ice loss in CMIP6 (Notz & Community, 2020) increases light availability in the upper ocean, thereby further acting to enhance phytoplankton growth rates and associated NPP. In addition to this intensified sea ice loss, a number of CMIP6 models include an improved representation of sea ice light attenuation, resulting in generally higher under-ice light transmission (Lebrun et al., 2023). Although CMIP6 simulations exhibit a larger decrease in upper Arctic Ocean nitrate concentrations over the 21$^{st}$ century (Fig. 3b), oligotrophic conditions are reached later or not at all, in contrast to CMIP5 models (Vancoppenolle et al., 2013), due to higher simulated mean state concentrations.

Although NPP is projected to increase in most regions of the Arctic Ocean, many models simulate local declines, particularly in Baffin Bay and the Nordic Seas. Regions experiencing NPP decline, such as Baffin Bay and the Nordic Seas, are characterized by rising temperatures, reduced nutrient availability and little or no sea ice in present-day conditions. As such, the potential for changes in light supply is low in these areas and phytoplankton growth shows little to no increase due to increased light availability. While higher temperatures do stimulate growth rates, the limiting effect of declining nutrient availability dominates, resulting in overall NPP declines in Baffin Bay and Nordic Seas. The projected decline in NPP in Baffin Bay is consistent with recent observations of decreasing productivity in this region (Ardyna & Arrigo, 2020)

Arctic Ocean phytoplankton communities are projected to shift toward smaller size classes, with miscellaneous phytoplankton increasingly dominating primary production throughout the 21st century (Fig. 9). This change in phytoplankton community structure aligns with previous projections from ESMs (Bopp et al., 2005; Fu et al., 2016), and is driven by differential nutrient affinities among PFTs. Miscellaneous phytoplankton demonstrate greater competitive advantage than diatoms under the increasingly low-nutrient conditions (Ward et al., 2012). In most of the Arctic Basin where NPP is increasing, productivity exhibits substantial gains, with miscellaneous phytoplankton driving most of this enhancement. This production is facilitated by newly ice-free waters that provide increased light availability and extended growing seasons. In contrast, in Baffin Bay and Nordic Seas, the decline of NPP is driven by a pronounced decrease in diatom productivity. While smaller miscellaneous phytoplankton can efficiently exploit low-nutrient environments and capitalize on higher temperatures, larger diatoms require higher nutrient concentrations to maintain their growth (Marinov et al., 2010). These contrasting patterns suggest a fundamental shift in Arctic marine ecosystems, with implications for higher trophic levels and thus carbon pump efficiency (Grebmeier et al., 2010; Ward et al., 2012).



This study highlights the complexity of projecting future NPP in the Arctic Ocean under anthropogenic pressure. While Arctic Ocean NPP is projected to increase throughout the 21st century, substantial uncertainties persist alongside pronounced spatial heterogeneity, with diverse implications. The projected shift in phytoplankton community structure toward smaller taxa will exacerbate the weakening of the Arctic Ocean carbon sink, which is already projected to decline

due to surface ocean warming that reduces $CO_2$ solubility and enhances stratification (Oziel et al., 2025). Concurrently, these ecosystem changes pose serious threats to local Inuit communities, whose economic livelihoods, food security, public health, and cultural identity depend heavily on marine resources (Malik & Ford, 2025; Mudryk et al., 2021). The magnitude and spatial distribution of future NPP increases remain highly uncertain due to poorly constrained key limiting factors. Specifically, uncertainties in light penetration through varying sea ice thickness and optical properties, coupled

with incomplete understanding of nutrient availability and cycling beneath ice-covered waters, represent critical knowledge gaps. Improving future Arctic Ocean NPP simulations requires enhanced representation of present-day nutrient levels, more accurate light transmission parameterizations through sea ice, and reduced climate sensitivity uncertainties across model ensembles.

## 5. Conclusion

The increase in Arctic Ocean NPP projected by CMIP6 models for the end of the 21st century is about four times higher than in CMIP5. This is mostly due to the stronger climate sensitivity of CMIP6 models, leading to greater warming and more rapid sea-ice decline. CMIP6 models also incidentally simulate higher nutrient levels than their CMIP5 counterparts. The NPP increase is accompanied by a shift in the phytoplankton community toward smaller species — more competitive under low-nutrient conditions — with impacts on higher trophic levels, biogeochemical cycling, and carbon export.

Implications for ecosystem services and Indigenous communities that depend on marine resources remain largely unassessed.

*Acknowledgements*

We gratefully acknowledge the ClimArctic project for supporting and enabling this study, as well as Olivier Torres for his help in processing the CMIP6 output data.

*Author contribution*

LK, MV and LCB conceived the study. LCB did the analysis and wrote the paper.

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
