# Peer review of "Heterogeneous future Arctic Ocean primary productivity changes projected in CMIP6"

_EGUsphere, 2025_

## Referee Comment (RC1)

**Review of *Hetrogeneous future of Arctic Ocean primary productivity changes projected in CMIP6* by Champiot-Bayard et al.**

**General Comments:**

This was a thoroughly interesting manuscript to read and - subject to some additional analysis and clarification - I would recommend its publication in BG. The authors set out to compare how model projections of phytoplankton growth and its drivers vary between CMIP5 and CMIP6 ensembles, how this varies spatially, and how this impacts different phytoplankton functional types with a projected shift to smaller species dominating. It was demonstrated that reduced light limitation is the primary driver of increased NPP in CMIP6 models, and a fourfold increase relative to that in CMIP5 is attributed to increased climate sensitivity in CMIP6 models. It was also noted that CMIP6 models simulate higher nutrient levels than in CMIP5 models, with oligotrophic conditions often not reached in the Arctic in CMIP6 models, unlike in CMIP5 projections.

My main concern is whether the result that CMIP6 models consistently remain above the oligotrophy threshold until 2100 is sensitive to the methodology used. Nutrient concentrations were averaged over the upper 100m of the ocean. It is unclear from the methods section what time period is being considered (annual means, monthly means, or something else), but depending on the time of year considered, this may be too deep. The Arctic upper mixed layer depth (MLD) is much shallower than this throughout much of the year (e.g. Peralta-Ferriz & Woodgate, 2015: https://doi.org/10.1016/j.pocean.2014.12.005; Allende et al., 2023: https://doi.org/10.1016/j.ocemod.2023.102226), and 100m is deeper than the annual maximum MLD in much of the Arctic. It could be the case that the nutrients are used up at the surface, but remain abundant below the upper mixed layer, leading to an average concentration above the oligotrophy threshold when nutrient concentrations are averaged over the top 100m, even if the surface does become oligotrophic. A robust justification of why 100m was chosen is required: an analysis of modelled MLDs throughout the CMIP6 ensemble would elucidate an appropriate depth threshold and allow for a meaningful discission of caveats in regions where the chosen threshold is less than optimal.

Primary productivity in the Arctic is characterised by a deep subsurface chlorophyl maximum but not all CMIP5 models reproduced this (e.g. Steiner et al., 2015: https://doi.org/10.1002/2015JC011232). To better understand how projections of Arctic Ocean primary productivity have evolved between CMIP5 and CMIP6, I would suggest that an analysis of vertical chlorophyll profiles should be included. As well as being an important companion to the horizontal validation presented in Figure 2, this would further justify the choice of depth over which to average the nitrate concentrations, strengthening the rest of the analysis.

Finally, while it is noted as a conclusion that CMIP6 models generally have higher nutrient concentrations than CMIP5, the increase in projected NPP (relative to the smaller increase in CMIP5) was attributed "mostly due to the stronger climate sensitivity of CMIP6 models, leading to greater warming and more rapid sea-ice decline". It is not clear to me how it was determined that the impact of stronger climate sensitivity than CMIP5 is more important than the role of less nutrient limitation compared to CMIP5. Expanding the discussion of the relative importance of each would be helpful to clarify this.

**Specific Comments:**

**P1 L35:** I think it's worth noting that there is disagreement on the sign of the projected changes in NPP across various regions in this paragraph.

**P2 L7:** Citation(s) for the enhanced precipitation, river inflow etc?

**P2 L9:** Cold you be more specific on the impacts that changes to the AMOC could have on the Arctic?

**P2 L22:** The implications of what extend beyond primary productivity, and how does this influence the region's role in global climate regulation? Possibly missing citation(s) here?

**P2 L24:** I would consider rephrasing this paragraph: it jumps from talking about recent changes in sea ice to a sentence on current nutrient supply and then back to projected changes in sea ice. Slightly reorganising this paragraph would improve overall readability.

**P2 L37:** Are you saying that temperature wasn't studied as a mechanism influencing NPP in Vancoppenolle et al. (2013) specifically, or more generally?

**P3 L20:** It would be useful to spell out what r1i1p1f1 means for readers less familiar with CMIP terminology.

**P3 L22:** There is a 5$^{th}$ scenario in Table 1: SSP5-3.4 that isn't mentioned in the four listed here. These seem like a reasonable spread of scenarios to consider, but is there a reason that no SSP4 scenario was analysed?

**P3 L34:** Is there a reason why only RCP8.5 was compared? The difference between some of the more optimistic CMIP5 and CMIP6 scenarios would also be interesting (but would arguably be beyond the scope of the paper.)

**P3 L31:** Is there a citation for the CDO remapdis software?

**P4 L4:** Pedantic point, but you haven't explicitly said that µ is the phytoplankton growth rate.

**P4 L8:** A reference is needed for the Arctic being (mostly) nitrogen limited. This is not necessarily the case everywhere in the Arctic though, particularly in the vicinity of Arctic rivers which are rich in nitrogen and silicic acid but poor in phosphate (e.g. Sakshaug, 2004; Popova et al., 2010). How would a consideration of non-nitrogen nutrient limitation affect the results?

**P4 L10:** Why was the top 100m chosen for averaging nitrate concentrations and how is the 1.6 mmol m$^{-3}$ oligotrophy threshold used in the analysis? From Figure 1 it seems to be annual means that are assessed against this threshold, but this isn't stated in the methods section. Given that nutrient concentrations exhibit strong seasonal variability, it would perhaps be more interesting to compare how the modelled seasonal cycle of nitrate concentrations is projected to change between present and end of century than only analysing the trend in annual means. (e.g. 1995-2014 nutrient seasonal cycle v 2081-2100 seasonal cycle)

**P6 L5:** This single sentence by itself seems unnecessary. Purely a stylistic choice, but I'd suggest removing it.

**P6 L8**: Does the primary production dataset used have a formal name?

**P6 L16:** Citation needed for "this depth range is most relevant to phytoplankton growth" (Links back to my main general comment).

**P7 L9:** The second sentence on the SST product doesn't give any specific information: including e.g. the spatial / temporal resolution and perhaps any issues with high-latitude satellite coverage would strengthen this sentence.

**Figure 1:** There is a lot of information presented in these figure, and some aesthetic changes would make it more easily interpretable:
- Recolour the ensemble members that are currently shown in black. Only use black for the mean to make it stand out more, and increase its thickness to match the red observations line.
- Consider using a different shade for each ensemble member so that, e.g. squares v diamonds isn't the only distinguishing feature between the two yellow lines.
- Adding year markers to panels a and b would help with interpretability. (This also goes for later figures where the year markers are only shown on figures in the bottom row.)
- Add a title to each subpanel stating which variable is being plotted.
- As noted in a previous comment, it is not clear from the methods section whether annual means or something else are being shown here (other than for SSIA); please clarify in the caption.

**P9 L10:** Can these observational uncertainties be quantified? (For all variables, but especially for $NO_3$ where the uncertainty is specifically mentioned.)

**Figure 2:** Consider using a different colourmap for panels a and b, perhaps white to green. I (and I expect other readers) would usually assume green to mean higher NPP, but here it's the other way round. An extra column showing model standard deviation would also be interesting: are the regions that show discrepancies with observations also regions where the models disagree with each other more strongly?

**Figure 3:** Elsewhere it is noted that CMIP6 models generally show more pronounced Arctic sea ice loss than CMIP5, but the decline in September Sea Ice Area is shown in Figure 3c is larger in CMIP5. Perhaps IFSD, as used in Figure 2, would be a more useful metric to show instead of SSIA?

**P11 L11:** I'm unsure what is meant by "larger lower sea ice extent", and 1881-1900 is not shown on Figure 3c.

**Figure 4:** Could you add information about where the models agree or disagree on the sign of the change, e.g. through hatching like in Figure 7 of Vancoppenolle et al. 2013? This would allow for a better comparison between the CMIP6 results presented here and the CMIP5 results shown there.

**P14 L1:** Missing a "the" before Laptev Sea.

**P15 L15:** Is the greater impact of nutrient concentrations for diatoms significant? The difference between diatoms and non-diatoms in Figure 7b is less than 1 standard deviation of multi-model uncertainty.

**Figure 8:** I would recommend using different colour scales for each variable; it is quite hard to discern any pattern from the $N_{lim}$ or $T_f$ figures with the current limits.

**Figure 9:** I would suggest making the subpanel labels more informative. E.g. "% of total NPP from diatoms in 1995-2014" for a; "% of total NPP from diatoms in 2081-2100" in b; "Change in % of total NPP from diatoms" for c. These are just suggestions, but it could be made clearer. It would also be better to be consistent with either "(2081-2100)-(1995-2014)" or "(2081-2100) relative to (1995-2014)". (Also there's a typo in the caption: € instead of (e).)

**Figure 10:** I know there are already 12 panels in this figure, but I think it would be better to include an extra row showing the distributions in 2081-2100 in addition to the baseline and anomalies already shown. For consistency, I think it would make more sense to use blue (as elsewhere in the paper) instead of pink to show a decline in the anomaly plots.

**P20 L17:** This relates to my main general comment: Figure 3 shows that annual means(?) of the mean upper 100m nitrate concentrations fail to reach oligotrophic conditions, but this doesn't necessarily mean that the upper ocean avoids oligotrophy closer to the surface.

**P20 L36:** It would be good to see a brief (sentence or two) overview of what the implications for higher trophic levels and carbon pump efficiency would be.

**P21 L16:** The stronger climate sensitivity and higher nutrient levels in CMIP6 compared to CMIP5 are both noted as conclusions, and higher projected NPP increase in CMIP6 than CMIP5 is attributed to the greater climate sensitivity rather than the higher nutrient levels. While Figure 11 shows that variables associated with increased climate sensitivity (reduced light limitation due to reduced sea ice, and $\Delta T_f$) drive the increase relative to historic levels in the CMIP6 projections, P19 L21 notes that some CMIP5 models are characterised by an initial increase in NPP followed by a decrease due to oligotrophy which isn't the case in CMIP6. It is unclear to me how the change relative to CMIP5 was attributed primarily to increased climate sensitivity rather than reduced nutrient limitation.

**P21 L19:** The impact on higher trophic levels is included as a conclusion, but it the discussion does not clearly say what the impacts would be.

**Additional References:**

Popova, E. E., Yool, A., Coward, A. C., Aksenov, Y. K., Alderson, S. G., de Cuevas, B. A., and Anderson, T. R.: Control of primary production in the Arctic by nutrients and light: insights from a high resolution ocean general circulation model, Biogeosciences, 7, 3569–3591, https://doi.org/10.5194/bg-7-3569-2010, 2010.

Sakshaug, E.: Primary and secondary production in the Arctic Seas, in: The Organic Carbon Cycle in the Arctic Ocean, edited by: Stein, R. and Macdonald, R., Springer, 57–81, 2004.